# Evidence for loss of contractile phenotype of the mouse aortic vascular smooth muscle (MOVAS) cell line with increasing number of passages *in vitro*

**Lucile Cadoret[1], Anaïs Okwieka[1], Alexandre Berquand[1], Christine Pietrement[1,2], Philippe Gillery[1,3], Stephane Jaisson[1,3]***

1 University of Reims Champagne-Ardenne, CNRS, Extracellular Matrix and Cell Dynamics unit (MEDyC) UMR, Reims, France, 2 University Hospital of Reims, Department of Pediatrics (Nephrology unit), Reims, France, 3 University Hospital of Reims, Department of Biochemistry-Pharmacology-Toxicology, Reims, France

* sjaisson@chu-reims.fr

## Abstract

### Aim

Vascular smooth muscle cells (VSMCs) are characterized by a considerable plasticity. Their phenotypic switch (from contractile to synthetic) plays a crucial role in the atherosclerotic process, explaining that numerous studies focus on this phenotypic transition. Thus, it is essential to use VSMCs that have been finely phenotyped for experimental purposes. The use of MOVAS cell line is suitable because, unlike primary cells, it is believed that these cells retain their phenotype and avoid cell senescence in culture. This study aimed to assess the phenotype of MOVAS cells over culture passages to ensure that they retained a contractile phenotype, before using them for further investigations.

### Methods

The phenotype of MOVAS cells at different culture passages (P3, P5 and P8) was analysed morphologically and by studying the expression of genes that indicate a contractile (*Acta2, Myocd* and *Cnn1*) and synthetic (*Klf4* and *Lgals3*) VSMC phenotype by RT-qPCR. Cell stiffness was analysed by atomic force microscopy and cell adhesion and migration.

### Results

Our results showed that MOVAS cells rapidly changed morphologically and that the gene expression of contractile markers was significantly reduced in favor of markers specific to the synthetic phenotype. These changes were associated with a reduction in cell stiffness and a significant increase in adhesion and migration properties.

**Data availability statement:** All relevant data are within the paper and its Supporting Information files.

**Funding:** L.C. received a grant from the Grand Est region (France) and the University of Reims Champagne Ardenne. This study was funded thanks to the Centre National de la Recherche Scientifique (CNRS), the University of Reims Champagne Ardenne and the Committee of American Memorial Hospital (Reims, France and Boston, MA, USA).

**Competing interests:** The authors have declared that no competing interests exist.

## Conclusion

MOVAS cells undergo a transition from contractile to synthetic phenotype with increasing number of passages *in vitro*, which means that these cells should be used with caution, at a low number of passages, while being regularly characterized.

## 1. Introduction

Vascular smooth muscle cells (VSMCs) play a crucial role in vascular homeostasis and in the pathogenesis of numerous cardiovascular diseases [1]. They are located in the media of blood vessel walls and are essential for regulating vascular tone and blood pressure, thanks to their contractile properties. VSMCs exhibiting this contractile phenotype in physiological conditions [2], are characterized by an elongated morphology and a contractile functional behavior related to the expression of specific biomarkers such as smooth muscle actin, calponin-1, myocardin or transgelin [3]. A striking characteristic of VSMCs is their plasticity and ability to transdifferentiate into different subtypes in response to alterations or stimuli from their microenvironment [4].

Atherosclerosis is a pathological process that causes many cardiovascular diseases, which incidence is constantly increasing all over the world, particularly in industrialised countries [5]. This process can affect young adults [6], even though ageing is recognized as a significant risk factor for its development [7]. It is characterised by the formation of atheroma plates located between intima and media of large and medium-sized arteries. This disease underlies many clinical events such as strokes, myocardial infarctions, or peripheral arterial diseases, since the plaques can limit or block local blood circulation, but also rupture and form a thrombus, thus obstructing blood flow and causing acute cardiovascular events. Early stages of atherosclerosis are characterized by the accumulation of oxidized low-density lipoprotein (LDL) particles in the subendothelium, forming initial lesions referred to as fatty streaks, which trigger an inflammatory response and attract circulating monocytes further differentiated into macrophages which phagocytize oxidized LDLs leading to the formation of foam cells [8]. VSMCs in a sclerotic artery dedifferentiate and migrate from the media to the subendothelium and contribute to the formation of a fibrous cap by synthesizing access fibrillar collagens [9]. Upon their dedifferentiation, VSMCs switch from a contractile to a synthetic phenotype characterized by morphological and gene expression changes. They adopt a star-shaped form and lose their specific contractile markers in favor of new markers such as galectin-3 or Krüppel-like factor 4 [10,11].

MOVAS (MOuse Vascular Aortic Smooth muscle) cells are primary VSMCs isolated from C57Bl6 mouse aortas and SV40-immortalized, and constitute a recognized cell model for studying vascular pathophysiology [12,13]. Originally isolated and characterized for their ability to retain the VSMC phenotypic characteristics, MOVAS cells are considered a valuable tool for exploring the underlying cellular mechanisms of vascular tone regulation, vessel repair, as well as progression of cardiovascular

diseases [14–17]. However, in order to support reliable and relevant studies, these cells must retain their contractile phenotype over passages in culture (as expected due to their immortalization). Indeed, any switch to a synthetic phenotype could alter the results, making the conclusions of the studies questionable.

Recent experiments using MOVAS cells carried out in our laboratory provided poorly reproducible results with cells from different passages, which led us to question the stability of their phenotype as a function of their *in vitro* ageing (*i.e.,* number of cell culture passages). The aim of the present study was then to characterize phenotypic differentiation profiles of MOVAS cells over cell culture passages *in vitro* in terms of morphology, function, and marker expression.

## 2. Materials and methods

### 2.1. MOVAS cells

The MOVAS cell line (CRL-2797™ - American Type Culture Collection (ATCC)) was cultured according to the supplier's recommendations, in Dulbecco's Modified Eagle's Medium (DMEM) 4.5 g/L glucose, containing 10% (v/v) fetal bovine serum (FBS) and 0.2 mg/L geneticin (G-418), at 37°C under 5% (v/v) $CO_2$ atmosphere. MOVAS cells were seeded at a density of $10^4$ cells/cm$^2$ and reached confluence in 3 or 4 days, which corresponded to a passaging frequency of every 4 days. The absence of mycoplasma contamination in cultures was regularly checked by PCR. In order to ensure reproducibility of the results, the present data are based on experiments carried out with different batches of cells but with the same lot number. Each vial of cells received from the ATCC was considered to be at passage 1 at the time of thawing, even though the exact number of cell culture passages was greater and varied between each batch of vial.

### 2.2. RT-qPCR

Total RNAs were obtained from MOVAS cells at P3, P5 and P8 using RNeasy Plus Kit (Qiagen – 74134) according to the manufacturer's instructions. Briefly, cells were lysed in a specific buffer containing guanidine-isothiocyanate and β-mercaptoethanol. Genomic DNA was removed by passing the lysate through a specific filtration column. Total RNAs were isolated by silica column extraction and reverse transcribed into cDNA using a Bio-Rad iScript kit (Bio-Rad) for 5 min at 25°C, 2 min at 46°C, and 1 min at 95°C. The samples were stored at −80°C until qPCR was performed using Bio-Rad iTaq Universal SYBR® Green Supermix (Bio-Rad – 1725121) on the CFX96™ Real-Time PCR System. The following thermocycling conditions were used: activation of polymerase at 95°C for 30 s, 39 cycles of denaturation at 95°C for 5 s, and elongation at 60°C or 62°C for 30 s. The expression of the genes of interest relative to housekeeping gene (*e.g., EEF1A1* gene) was calculated using the Pfaffl method [18]. Details of the sequences and primer hybridization temperatures, as well as PCR product lengths, are available in the supplemental data file.

### 2.3. Actin labelling and analysis of cell morphology

7,500 MOVAS cells at P3, P5 and P8 were seeded in 8-chamber LabTek wells (Nunc Lab-Tek II Chamber Slide System, Thermo Fisher Scientific). After adhesion, cells were rinsed with Dulbecco's Phosphate Buffered Saline (DPBS) and incubated at 37°C in a 5% (v/v) $CO_2$ atmosphere for 48 h in DMEM 4.5g/L glucose without FBS. After 48 h of incubation, cells were stained as described below. After rinsing with 100 μL DPBS, cells were fixed with 4% (v/v) paraformaldehyde for 10 min and washed twice with DPBS. A solution of 0.1% (v/v) Triton-X100 was used to permeabilize cells for 10 min at room temperature (20°C). Cells were then incubated with 3% (m/v) BSA for 2 h to prevent non-specific antibody binding. Finally, cells were labelled in the dark with Actin-555-Red (Molecular Probes - R37112) for 30 min at room temperature, and cellular DNA was stained with DAPI (4′,6-diamidino-2-phenylindole, Southern Biotech – 0100–20). Actin probe was used according to the manufacturer's instructions (2 drops diluted per ml of DPBS), while DAPI was already present in the mounting medium (no information from the manufacturer regarding its concentration). supplier instructions (Immunolabelled samples were visualized using a fluorescence microscope equipped with the

appropriate filters to detect Actin-555 and DAPI signals. Images were captured under excitation (540 nm and 355 nm, respectively) and emission (565 nm and 405 nm, respectively) wavelengths adapted to each fluorochrome. The images were digitally reprocessed to convert the red pixels to green to make the images more visually appealing. The reported actin fluorescence represents the average fluorescence intensity within each cell for all cells of the field from randomly taken images. The ImageJ plugin MorphoLibJ, created at the INRA-IJPB digital imaging and modeling laboratory was used to study cell morphology [19]. More details about the parameters used for the characterization of cell morphology are provided in supplemental data file.

## 2.4. Atomic force microscopy (AFM) imaging

300,000 MOVAS cells at P3, P5 and P8 in 1.5 mL of culture medium were seeded in glass-cups (WillCo - HBST-3512). After adhesion, cells were rinsed with DPBS, serum-starved for 24 h, and incubated at 37°C under 5% (v/v) $CO_2$ atmosphere for 48 h in DMEM 4.5g/L glucose. After rinsing with 1 mL of DPBS, 2 mL of fresh culture medium were added, and the cup was positioned on the atomic force microscope. To characterize the mechanical properties of the cells, the PFT-QNM (PeakForce Tapping – Quantitative Nanoscale Mechanical Characterization) mode was used. Cells were imaged with ScanAsyst-FLUID probes (Bruker, Billerica, USA) on a Bioscope CatalystTM/Nanoscope 8.15 version (Bruker Billerica, USA), with the following parameters: scan rate 0.3 Hz, resolution 256 pixels/line, PeakForce Amplitude 2000 nm and average force of 300 pN. A total of 65,536 force curves were analyzed for each picture. Young's modulus (YM) values were measured on individual cells and each point on the graph corresponds to the average value obtained from 5,184 measurements per cell realized in a defined and calibrated area (36x36 = 1296 pixels, 4 force curves per pixel).

More details about AFM calibration as well as mapping and calculation of YM values are available in supplemental data file.

## 2.5. Cell adhesion assay

20,000 MOVAS cells at P3, P5 and P8 were seeded per well with 500 µL of serum-free DMEM 4.5g/L glucose in a 24-well plate and incubated at 37°C in a 5% (v/v) $CO_2$ atmosphere. After 2h of incubation, cells were rinsed with 200 µL DPBS once and fixed for 20 min at room temperature (20°C) using a 1.1% (v/v) glutaraldehyde solution prepared in DPBS. Cells were then stained with 0.1% (m/v) crystal violet for 20 min in the dark. After rinsing and drying, 500 µL of 10% (v/v) acetic acid were added per well to elute the dye. Absorbances were then measured spectrophotometrically at 560 nm.

## 2.6. Proliferation assay

Cell proliferation was assessed by the crystal violet coloration method. 20,000 cells were seeded per well in a 24-well plate and incubated at 37°C in a 5% (v/v) $CO_2$ atmosphere in DMEM 4.5g/L glucose with 10% (v/v) FBS. At days 1, 3, and 6, MOVAS cells of each passage were fixed using a 1.1% (v/v) glutaraldehyde solution prepared in DPBS at room temperature (20°C) for 20 min. After rinsing with DPBS, cells were stained with crystal violet for 20 min in the dark. After rinsing and drying, 500 µL of 10% (v/v) acetic acid was added per well to elute the dye. Absorbances were then measured spectrophotometrically at 560 nm.

## 2.7. Migration assay

Cell migration was determined using Ibidi® system (Ibidi – 80209). MOVAS cells at P3, P5 and P8 were plated at a density of 7,000 cells/chamber in a two-chamber insert. After 24 hours, the Ibidi inserts were removed to create a gap, floating cells were eliminated by a DPBS rinse, and the remaining cells were cultured in DMEM 4.5g/L glucose without FBS at 37°C in a humidified incubator under a 5% (v/v) $CO_2$ atmosphere. At 0, 24, and 48 hours, migration was observed by taking photographs using a microscope at x10 magnification and by measuring the progress of gap closure. The size of the gap area allowing to indirectly reflect cell migration was measured using ImageJ software.

## 2.8. Statistical analyses

GraphPad Prism 8.0. software was used to analyze data and generate graphs. Before statistical comparison, D'Agostino & Pearson's normality test has been applied to our data showing anormal distribution in somes cases. For that reason, non-parametric ANOVA (Kruskal-Wallis test) was used to compare data between different passages and Mann-Whitney's U test was used for direct comparisons between P3 and P5, P5 and P8, P3 and P8. All data are represented as means±standard deviations. Statistical tests used and n-numbers in each experiment are indicated in the corresponding figure legends.

## 3. Results

### 3.1. MOVAS cell morphology changes with increasing number of passages

Cell morphology was analyzed by quantifying various parameters such as area, perimeter, and lengthening (details for these measurements are available in methods section and in supplemental data file). Fig 1A shows the appearance of the MOVAS cells at different culture passages. At P3, the cells exhibited a typical appearance of contractile VSMCs, with an elongated, fusiform shape. They progressively lost this appearance after successive passages, showing a very different morphology at P8, as assessed by various parameters (Fig 1B). For example, average cell area was 12,902±2,835 µm² at P3, 9,352±2,129 µm² at P5 and 7,251±2,850 µm² at P8, showing a significant decrease of −28% and −44%, respectively. Cells also appeared less elongated at P5 and P8 (−59% for both conditions, $p < 0.001$). However, cell perimeter

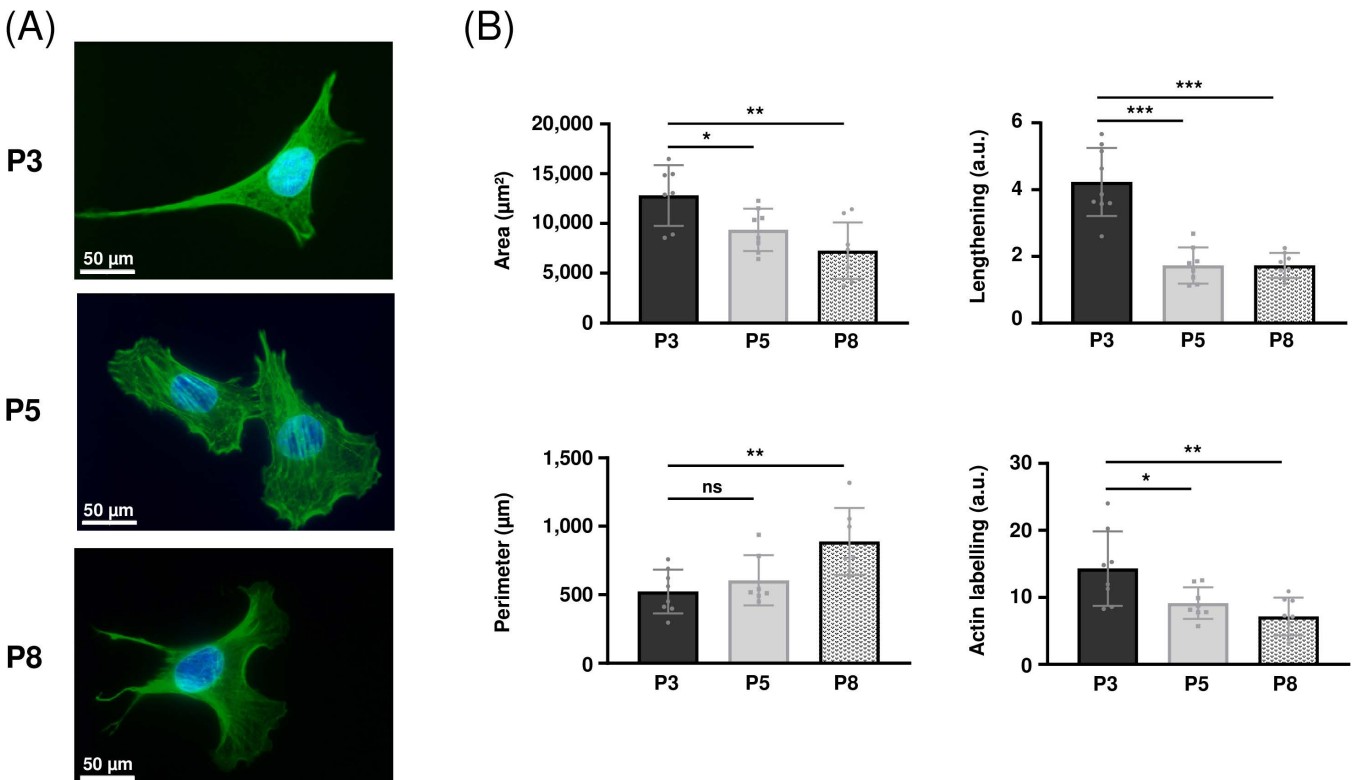

**Fig 1. Changes in morphology of MOVAS cells with increasing number of passages.** (A), MOVAS cells at different passages (P3, P5 and P8) after actin (green) and nuclei (DAPI, blue) staining. Magnification x63, scale bar: 50 µm. (B), Quantification of parameters assessing cell morphology (area, lengthening, perimeter) and of actin expression, calculated using the ImageJ MorpholibJ plugin from cell pictures (n=8). ***:$p < 0.001$, **:$p < 0.01$,*:$p < 0.05$, ns: not significant. a.u.: arbitrary units.

gradually increased from 524±160 µm at P3 to 900±230 µm at P8 (+72%, p<0.001), showing more complex stellar shapes. These data were completed by quantification of actin labelling, which showed a clear decrease of actin content between P3 and P5 (14.3±5.5 vs 9.1±2.4 a.u., −36%, p<0.05) and between P3 and P8 (14.3±5.5 vs 7.2±2.8 a.u., −50%, p<0.01).

### 3.2. Loss of contractile phenotype with increasing number of passages

To confirm the potential phenotypical switch suggested by the changes in cell morphology, gene expressions of characteristic markers of contractile and synthetic phenotypes were analyzed by RT-qPCR. The following contractile markers were analyzed: α2 smooth muscle actin (*Acta2* gene), calponin-1 (*Cnn1*) and myocardin (*Myocd*), whereas synthetic markers were the transcription factor Krüppel-like factor 4 (*Klf4*) and galectin-3 (*Lgals3*). *EEF1A1* gene was used as reference gene and no variation of expression between different passages was observed (data not shown).

Expression of contractile markers was significantly decreased in P5 and P8 MOVAS cells in comparison with P3 ones (Fig 2A). Indeed, *Acta2* expression was respectively 39% and 43% lower at P5 and P8 (p<0.001). The same expression profile was obtained for *Cnn1* and *Myocd* genes: expression levels were respectively 74% and 40% lower at P8 cells in comparison with P3 (p<0.001). In contrast, MOVAS cells exhibited an increased expression of synthetic phenotype markers with increasing numbers of passages (Fig 2B). *Klf4* expression was significantly higher at P5 (+130%, p<0.001) and P8 (+131%, p<0.001) in comparison with P3. A similar profile was observed for *Lgals3* (+39% and +82%, p<0.001, at P5 and P8, respectively). These results are summarized in Fig 2C which clearly shows a decreased expression of contractile

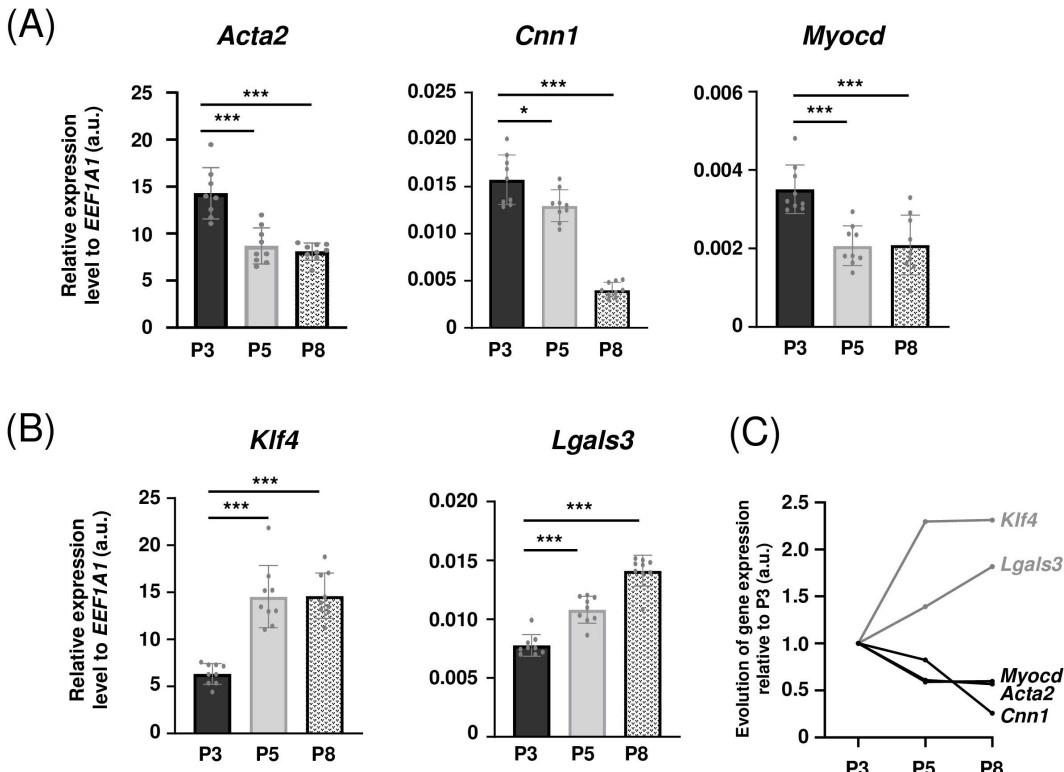

**Fig 2. Contractile to synthetic phenotype switch of MOVAS cells with increasing number of passages.** RT-qPCR analysis of gene expression of contractile (A, *Acta2*, *Myocd*, *Cnn1*) and synthetic (B, *Klf4*, *Lgals3*) phenotype markers in MOVAS cells at P3, P5, and P8 (n=9). Evolution of relative expression of each markers was presented in panel C. *EEF1A1* gene was used as reference gene. ***:p<0.001, *:p<0.05. a.u.: arbitrary units.

markers (black lines) alongside an increase in synthetic phenotype markers (grey lines), and are in addition confirmed by inverse correlations established between contractile and synthetic markers (S1 Fig).

### 3.3. Decreased cellular stiffness with increasing number of passages

Phenotypical changes being usually associated with modifications in mechanical properties, cells were then analyzed by atomic force microscopy (AFM). Colors and textures in the topography images (Fig 3A) illustrate modifications in cell architecture, especially regarding the cytoskeleton actin fibers which became less visible as the number of passages increased. Young's Modulus (YM) measurements allowing to evaluate cell stiffness were taken over the surface of the cells (Fig 3B). YM values shifted towards lower values as the number of passages increased, with significant ($p < 0.01$) decreases from $61.1 \pm 32.4$ kPa for P3 to $31.7 \pm 14.7$ kPa (P5) and $30.6 \pm 6.8$ kPa (P8). These results are in favor of a decreased stiffness of cells after several *in vitro* culture passages.

### 3.4. Increased adhesion and migration ability with increasing number of passages

Following the demonstration of a phenotypic change of MOVAS cells as the number of culture passages progressed, several of their properties (*i.e.,* adhesion, proliferation and migration) were tested in order to evaluate the impact of the phenotypic change on the behavior of these cells. The adhesion assay showed no significant difference between P3 and P5 cells after 2 hours of incubation (Fig 4A). However, P8 cells showed a significant increase (3-fold, $p < 0.001$) of adhesion, with absorbance values rising from $0.071 \pm 0.012$ (P3) to $0.210 \pm 0.010$ (P8). In the proliferation test, which was carried out over 6 days at 37°C in the presence of serum-containing medium, the proliferation kinetics of MOVAS cells were not statistically different, regardless of the number of passages (Fig 4B). Finally, the assessment of cell migration ability showed that P3 cells exhibited a low percentage of gap closure and therefore a low migration potential at 24 h ($6.9 \pm 4.9\%$) and 48 h ($7.2 \pm 4.3\%$), which was in agreement with their contractile phenotype (Fig 4C). While P5 cells showed a similar percentage of gap closure to P3 cells at 24 h of incubation, a significant increase ($28.4 \pm 9.4\%$, $p < 0.001$) was observed at 48 h. In contrast, P8 cells showed a significant increase in migration at 24 h ($30.9 \pm 12.6\%$, $p < 0.001$), which remained of the same order of magnitude at 48 h ($36.0 \pm 14.4\%$, $p < 0.001$).

## 4. Discussion

Numerous studies have shown that phenotypic changes in VSMCs play a major role in the atherosclerotic process [9]. These cells exhibit a considerable plasticity and are able, under the effect of different stimuli, to dedifferentiate into numerous subtypes, ranging from macrophage-like cells to osteoblast-like [4]. Without going as far as total dedifferentiation, they can also display more subtle changes in their behavior, moving from a contractile to a synthetic or secretory phenotype [20]. This phenotypic switch has been investigated for several years, as it is considered to be one of the first steps leading to the migration of VSMCs from the media to the subendothelial space [21,22].

Our team is taking part in a research project aiming at studying the impact of the molecular modification of the vascular matrix microenvironment on cell behavior [23,24], and specifically on this phenotypic switch. For that purpose, it is essential to use well-characterized cells whose contractile phenotype is unquestioned, and which must be maintained over culture passages in order to guarantee the reproducibility and robustness of experimental results. Indeed, cells can undergo senescence *in vitro* and dedifferentiate over passages [25]. VSMCs from primary cultures are notorious for becoming senescent and rapidly losing their morphology after a limited number of passages, even just after isolation from the original vessel suggesting that the detachment from neighbouring cells seems to be a powerful trigger for phenotype switching [26]. The same phenomenon has been observed with human mesenchymal stem cells [27,28].

To overcome this problem, we decided to use a mouse aortic VSMC line, *i.e.,* MOVAS cells, since SV40 immortalized cell lines are known to retain their phenotype in culture, thus guaranteeing reproducible results [13]. However, after characterizing the initial contractile phenotype of these MOVAS cells, we noted a change in morphology and behavior over

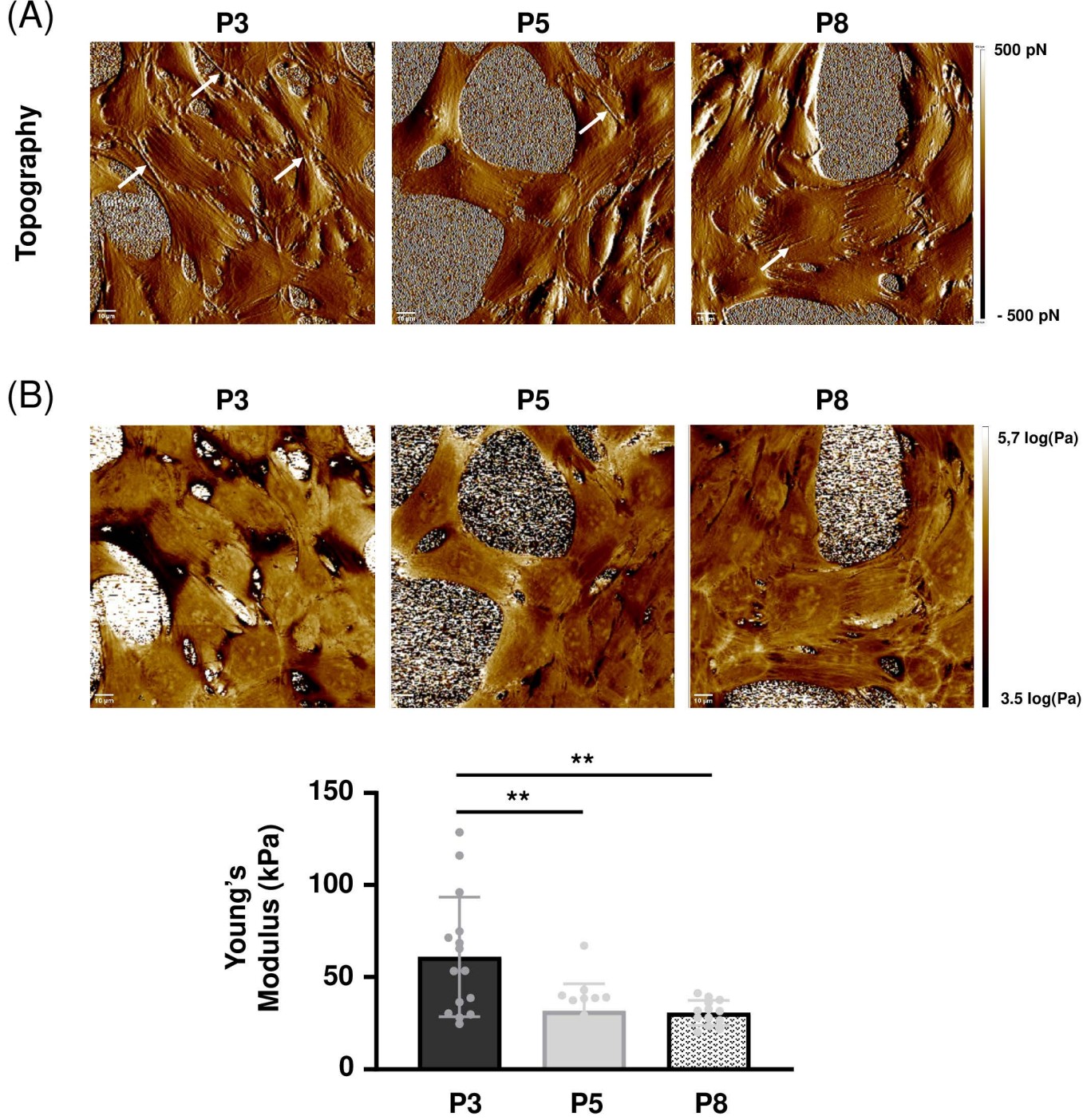

**Fig 3. Decreased stiffness of MOVAS cells with increasing number of passages.** MOVAS cells at P3, P5 and P8 were analyzed by atomic force microscopy using PFT-QNM mode. (A) Peak force error images (scale: −500 to 500 pN) showing cell topography. Arrows show cytoskeleton actin fibers (B) Mapping of Young's modulus obtained for cells at P3, P5 and P8. Young's modulus values were measured on individual cells and each point on the graph corresponds to the average value obtained from 5,184 measurements per cell (n = 15). **:p < 0.01. Scale bar: 10μm.

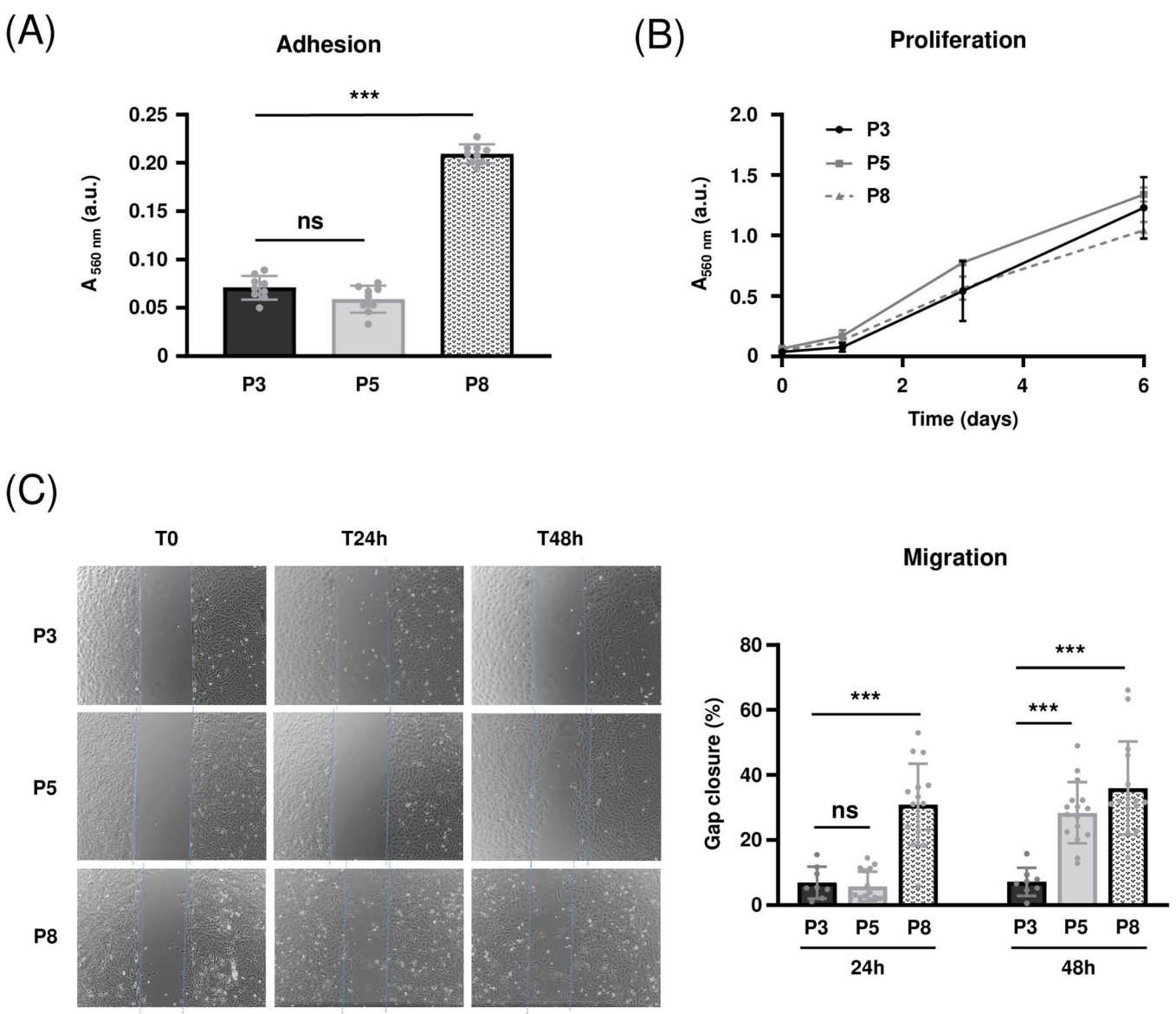

**Fig 4. Increased adhesion and migration ability of MOVAS cells with increasing number of passages.** (A) Adhesion: 20,000 MOVAS cells at P3, P5, and P8 were seeded per well in plastic a 24-well plate in serum-free medium and incubated for 2h at 37°C. Adherent cells were stained with crystal violet and quantified by measurement of absorbance at 560 nm (n = 9). (B) Proliferation: 20,000 MOVAS cells were seeded per well in a 24-well plate in culture medium containing 10% (v/v) serum, and incubated for 1, 3 and 6 days at 37°C. At each incubation time, cells were stained with crystal violet and quantified by measurement of absorbance at 560 nm (n = 18). (C) Migration: MOVAS cells were plated at a density of 7,000 cells/chamber in a two-chamber Ibidi® insert. After cell adhesion, inserts were removed and cells incubated in serum-free medium for 48h. Pictures were taken over time to monitor gap closure progress (on the left) and cell-free area allowing to indirectly reflect cell migration was measured using ImageJ software (n = 16). ***: p < 0.001, ns: not significant. a.u.: arbitrary units.

culture passages, which was suspected to be the source of a large variability in the experimental results. Therefore, the aim of this study was to characterize in more detail the evolution of the phenotype and behavior of MOVAS cells in culture.

The first sign that drew our attention was the change in cell morphology over culture passages. Initially, MOVAS cells were elongated and spindle-shaped, characteristic of the contractile phenotype [12], but they rapidly evolved towards a

more rhomboid and star-shaped shape. Analysis of cell morphology using appropriate software enabled us to objectively characterize and demonstrate this phenotypic change, showing, for example, a significant reduction in cell lengthening associated with an increase in shape complexity (through an increase of their perimeter). Similar morphological switches attesting VSMC dedifferentiation have already been described in the literature [29,30], for example in a study designed to investigate the effect of rapamycin on this phenotypic switch [2] or when VSMCs are cultured in the presence of advanced glycation end-products [31]. Changes in the actin cytoskeleton have also been demonstrated in aging VSMCs [32], supporting our observations.

To confirm this phenotypic change, the expression of various markers specific to each phenotype was studied at P3, P5 and P8, corresponding to the passages at which the morphological changes were observed. It was therefore decided to study the expression of the *Acta2* [2,33], *Cnn1* [1,34] and *Myocd* [35] genes, specific to the contractile phenotype, and the *Lgals3* [36,37] and *Klf4* [38,39] genes for the synthetic and secretory phenotype. The results showed a significant decrease in the expression of contractile markers from P3 to P8, with a corresponding increase in markers characteristic of the proliferative phenotype. Such changes in the gene expression profile are reported to be associated with a phenotypic switch of VSMCs [3,9,30,40], which confirms the phenotypic switch of MOVAS cells in the culture conditions recommended by the supplier.

As changes in cell morphology and in the organization of actin cytoskeleton may be associated with changes in cellular stiffness [41], the MOVAS cells at different passages were analyzed by AFM. YM measurements on VSMCs (in cultures or *in situ* within a vessel) have already been carried out in the literature [32,42]. The YM values obtained in the present study are compatible with those described elsewhere and show a decrease in YM over culture passages resulting in a decrease in MOVAS cell stiffness, probably explained by the phenotypic switch. These variations in YM were in accordance with those showed in the study by Qiu *et al.* [32], VSMCs from old monkeys showing a lower stiffness.

Finally, given that the phenotypic changes are always associated with modifications in cell behavior characterized by an increase in the migratory and proliferative capacities of VSMCs [43,44], these properties were evaluated on MOVAS cells at different passages. Our results clearly demonstrate an increase in cell adhesion and migration at P8 compared with P3, while proliferation over 6 days remains unchanged. These results corroborate the phenotypic switch observed previously in terms of changes in marker expressions, although it is difficult to explain why no changes in proliferative properties were observed. However, one explanation could be the use of culture medium supplemented with serum for proliferative assays whereas the other assays were performed using serum-free medium.

A major limitation of this study is that we used a single batch of MOVAS cells, even though the experiments were reproduced using different vials from this batch. As it is not possible to obtain precise information on how many times the cells were passed prior to being frozen and sold by the supplier, the number of passages mentioned in this article (P3 to P8) can only apply to this particular batch of cells and cannot be extrapolated to other batches.

## 5. Conclusion

In order to enable reliable interpretation of the results of *in vitro* studies of the molecular mechanisms involved in the atherosclerotic process, it is essential to have knowledge of the phenotype of the vascular smooth muscle cells (VSMCs) used for these studies, which requires detailed phenotypic characterization. Using cell lines such as MOVAS cells was thought to guarantee this phenotypic stability. However, our study's results show that the MOVAS cells undergo premature phenotypic change as early as P5, losing their contractile phenotype in favour of a synthetic one. It would now be appropriate to compare the expression of different markers in MOVAS cells with that in freshly isolated primary VSMCs, to ensure that their phenotypes are quite similar. If this is the case, using these cells at an early passage should produce reliable results in *in vitro* experiments exploring the mechanisms involved in the atherosclerotic process.

However, the key lesson to be learned from this study is that precautions should be taken when using this cell line, particularly through regular phenotypic characterization.

## Supporting information

**S1 File. Supplemental data.**
(DOCX)

**S1 Table. Sequences of primers used for RT-qPCR analyses.**
(DOCX)

**S1 Fig. Relative expression of contractile vs synthetic markers in MOVAS cells.** RT-qPCR analysis of gene expression of contractile (Acta2, Cnn1, Myocd) vs synthetic (Lgals3, Klf4) phenotype markers in MOVAS cells at different passages were pooled in the same plot. Regression lines were calculated and a Pearson correlation test was used to determine the statistical significance and the r coefficient (indicated within each graph). The regression lines are represented by black lines, with 95% confidence intervals shown in grey.
(DOCX)

## Acknowledgments

The authors thank the NanoMat/URCATech platform, which enabled the AFM analyses to be carried out. L.C. received a grant from the Grand Est region (France) and the University of Reims Champagne Ardenne. This study was funded thanks to the *Centre National de la Recherche Scientifique* (CNRS), the University of Reims Champagne Ardenne and the Committee of American Memorial Hospital (Reims, France and Boston, MA, USA).

## Author contributions

**Conceptualization:** Philippe GILLERY, Stephane JAISSON.

**Data curation:** Lucile CADORET, Anaïs OKWIEKA, Alexandre BERQUAND.

**Formal analysis:** Alexandre BERQUAND.

**Funding acquisition:** Christine PIETREMENT, Stephane JAISSON.

**Investigation:** Lucile CADORET, Anaïs OKWIEKA, Alexandre BERQUAND, Christine PIETREMENT.

**Methodology:** Christine PIETREMENT.

**Supervision:** Christine PIETREMENT, Philippe GILLERY, Stephane JAISSON.

**Validation:** Stephane JAISSON.

**Writing – original draft:** Lucile CADORET, Stephane JAISSON.

**Writing – review & editing:** Christine PIETREMENT, Philippe GILLERY, Stephane JAISSON.

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
