## [Decision Letter · Decision Letter 0]

27 Jul 2025

Dear Dr. Jaisson,

Thank you very much for submitting your manuscript to PLOS ONE. After careful consideration, we have determined that your manuscript has the potential to be published, although some minor aspects require attention. Indeed, all three reviewers have raised points that require clarification. We therefore kindly ask you to provide clear and comprehensive responses to their comments.

We look forward to receiving your revised manuscript.

Kind regards,

Gianfranco Pintus, MSc, PhD.

Academic Editor

PLOS ONE

Journal Requirements:

The authors thank the NanoMat/URCATech platform, which enabled the AFM analyses to be carried out. L.C. received a grant from the Grand Est region (France) and the University of Reims Champagne Ardenne. This study was funded thanks to the Centre National de la Recherche Scientifique (CNRS), the University of Reims Champagne Ardenne and the Committee of American Memorial Hospital (Reims, France and Boston, MA, USA).

L.C. received a grant from the Grand Est region (France) and the University of Reims Champagne Ardenne. This study was funded thanks to the Centre National de la Recherche Scientifique (CNRS), the University of Reims Champagne Ardenne and the Committee of American Memorial Hospital (Reims, France and Boston, MA, USA).

Reviewers' comments:

Reviewer's Responses to Questions

**Comments to the Author**

1. Is the manuscript technically sound, and do the data support the conclusions?

Reviewer #1: Yes

Reviewer #2: Yes

Reviewer #3: Yes

2. Has the statistical analysis been performed appropriately and rigorously?

Reviewer #1: I Don't Know

Reviewer #2: Yes

Reviewer #3: I Don't Know

3. Have the authors made all data underlying the findings in their manuscript fully available?

Reviewer #1: Yes

Reviewer #2: Yes

Reviewer #3: No

4. Is the manuscript presented in an intelligible fashion and written in standard English?

Reviewer #1: Yes

Reviewer #2: Yes

Reviewer #3: Yes

Reviewer #1: The manuscript titled "Evidence for Loss of Contractile Phenotype of the Mouse Aortic Vascular Smooth Muscle (MOVAS) Cell Line with Increasing Number of Passages In Vitro" is well-structured and logically presented. It addresses a critical aspect of vascular smooth muscle cell (VSMC) research, emphasizing the importance of monitoring cell phenotype in vitro. The data is robust, and the conclusions are well-supported.

To improve the manuscript, ensure consistent use of abbreviations (e.g., VSMCs, MOVAS) and correct minor grammatical and typographical errors. Including a brief mention of future research directions in the conclusion would provide a broader perspective. Add background information on the common use and advantages of MOVAS cells over primary VSMCs and explain the significance of maintaining the contractile phenotype for cardiovascular research. Clearly state the hypothesis or research question and describe the statistical methods used, including specific tests and significance criteria.

Expand the discussion to compare findings with other studies and place them in the broader context of VSMC research. Discuss potential study limitations and suggest how they could be addressed in future research. Highlight specific applications or implications for cardiovascular disease research or treatment. Provide recommendations for researchers using MOVAS cells, ensure all references are current and relevant, and consider adding more references to support key points. Include details on Atomic Force Microscopy (AFM) calibration and settings for Young's modulus mapping. Describe statistical tests in the methods section, detailing the number of replicates and sample sizes. Clarify the rationale for choosing non-parametric ANOVA and Mann-Whitney's U test, ensuring their appropriateness given the data distribution.

Reviewer #2: I believe the manuscript is a very straight forward study that clearly shows evidence for the phenotypic shift of the Movas cells with advancing passage. I think the phrase "phenotypically stable" to describe the overall history of the cell line in the Introduction may be too strong though and should be avoided. It may be more accurate to mention a few studies, such as with calcification where they did share phenotype with primary cells. In addition, this cell line was immortalized with viral methodology which altered other phenotype changes such as loss of senescence.

Reviewer #3: Dear authors,

thank you for your submission. Your study adds important knowledge to this field and is of special interest to research performing basic studies on MOVAS.

However, with the limited information you provide in your methods section most experiments seem difficult to reproduce (e.g., staring with the frequency of passaging of your cells). In addition a few things are missing to make your results and conclusions more reliable and scientifically sound.

For your statistics:

- power calculation (with the significance you show I have no doubts that your n-number is in an adequate range, yet a power calculation should be performed prior to your experiments)

- correlation analysis between the contractile and synthetic gene expression levels > proof of a reverse correlation may support the hypothesis of a switch from contractile to synthetic over time

For your methods/results:

- staining protocol for figure 1 A is missing and selection of images might not be ideal

- stability of your reference gene expression over the different passages must be tested

- product length of PCR products missing in suppl. table, so does the inclusion of any internal PCR controls

For your discussion:

- ideally find studies that used MOVAS to study atherosclerosis specifically

Please also have a look at the attached word document that provides more detailed explanations in the comments.

Thank you and warmest regards.

**Do you want your identity to be public for this peer review?** For information about this choice, including consent withdrawal, please see our Privacy Policy

Reviewer #1: No

Reviewer #2: No

Reviewer #3: No

---

## [Author Response · Author response to Decision Letter 1]

4 Sep 2025

Please see the "Response to reviewers" file

---

## [Decision Letter · Decision Letter 1]

1 Oct 2025

Thank you very much for submitting your manuscript to PLOS ONE. After careful consideration, we have concluded that your manuscript has the potential to be published, although some minor aspects need to be addressed. In particular, we ask that you provide a response to the questions raised by Reviewer 3. We therefore invite you to revise your manuscript, paying special attention to the points indicated by this reviewer.

We look forward to receiving your revised manuscript.

Kind regards,

Gianfranco Pintus, MSc, PhD.

Academic Editor

PLOS ONE

Journal Requirements:

Reviewers' comments:

Reviewer's Responses to Questions

**Comments to the Author**

Reviewer #1: All comments have been addressed

Reviewer #2: All comments have been addressed

Reviewer #3: (No Response)

2. Is the manuscript technically sound, and do the data support the conclusions?

Reviewer #1: Yes

Reviewer #2: Yes

Reviewer #3: Yes

3. Has the statistical analysis been performed appropriately and rigorously?

Reviewer #1: Yes

Reviewer #2: Yes

Reviewer #3: No

4. Have the authors made all data underlying the findings in their manuscript fully available?

Reviewer #1: Yes

Reviewer #2: Yes

Reviewer #3: Yes

5. Is the manuscript presented in an intelligible fashion and written in standard English?

Reviewer #1: Yes

Reviewer #2: Yes

Reviewer #3: Yes

Reviewer #1: It is a nice study about mouse arterial smooth muscle cell proliferation. I have no further comments

Reviewer #2: (No Response)

Reviewer #3: Abstract:

no comments> all suggestions actioned adequately

Introduction:

no comments> all suggestions actioned adequately

Materials & Methods:

no comments > all suggestions actioned adequately

Results:

> PCR results:

Addition to Figure 2C: I appreciate that you decided to show the progression of your genes of interest over the different passages.

However, proof of linear correlation requires a statistical test (to my knowledge the standard test is the Pearson Correlation test that allows you to calculate an R-value). I suggest you add the calculated R value and the correlation graph to the supplements and to comment on the correlation in the figure legend of 2C.

Discussion and Conclusion:

No comments > all suggestions actioned adequately

**Do you want your identity to be public for this peer review?** For information about this choice, including consent withdrawal, please see our Privacy Policy

Reviewer #1: No

Reviewer #2: No

Reviewer #3: No

---

## [Author Response · Author response to Decision Letter 2]

13 Oct 2025

PONE-D-25-33029R1

Cadoret et al. - Evidence for loss of contractile phenotype of the mouse aortic vascular smooth muscle (MOVAS) cell line with increasing number of passages in vitro.

Responses to reviewers:

Reviewer #3:

Results:

> PCR results:

Addition to Figure 2C: I appreciate that you decided to show the progression of your genes of interest over the different passages.

However, proof of linear correlation requires a statistical test (to my knowledge the standard test is the Pearson Correlation test that allows you to calculate an R-value). I suggest you add the calculated R value and the correlation graph to the supplements and to comment on the correlation in the figure legend of 2C.

Answer: As suggested by Reviewer #3, regression lines and Pearson correlation coefficients were calculated and presented in supplemental data file (Fig. S1). R-values were also mentioned in the main text (page 10).

---

## [Decision Letter · Decision Letter 2]

25 Oct 2025

We look forward to receiving your revised manuscript.

Kind regards,

Gianfranco Pintus, MSc, PhD.

Academic Editor

PLOS ONE

Journal Requirements:

Additional Editor Comments:

Thank you for submitting your manuscript to PLOS ONE. After careful consideration, we continue to see merit in the work. However, one of the reviewers still requests revisions. From our reading of the exchange, it appears there was a misunderstanding of the reviewer’s original request. In the editor’s view, the requested changes are not strictly essential for publication, but addressing them would help consolidate and strengthen the manuscript’s conclusions. We therefore invite you to submit a revised version that clarifies and directly addresses the reviewer’s points, with a clear, point-by-point response and, as appropriate, corresponding changes highlighted in the text.

Reviewer's Responses to Questions

**Comments to the Author**

Reviewer #3: (No Response)

2. Is the manuscript technically sound, and do the data support the conclusions?

Reviewer #3: Yes

3. Has the statistical analysis been performed appropriately and rigorously?

Reviewer #3: No

4. Have the authors made all data underlying the findings in their manuscript fully available?

Reviewer #3: Yes

5. Is the manuscript presented in an intelligible fashion and written in standard English?

Reviewer #3: Yes

Reviewer #3: Dear authors,

Thank you for resubmitting your manuscript. I apologize for my probably vague instructions regarding the correlation analysis last time but it requires adaptation. I would suggest you to consider seeking support from a biostatistician.

What you need to correlate is the relative expression levels of the genes that indicate a contractile phenotype with the relative gene expression levels of the genes that indicate a more synthetic phenotype. You basically do this to support the conclusion that the expression of contractile genes decreases whilst at the same time the expression of synthetic genes increases and to answer the question if their expression is somewhat connected/correlated with one another.

You may require a specific correlation test to do that, as your expression levels might be inversely correlated.

If you can show that it would be a suppotive and strong statistical evidence for your conclusions.

I hope this was helpful.

Best regards.

The reviewer.

**Do you want your identity to be public for this peer review?** For information about this choice, including consent withdrawal, please see our Privacy Policy

Reviewer #3: No

---

## [Author Response · Author response to Decision Letter 3]

12 Nov 2025

Responses to reviewers:

Reviewer #3:

Results:

Dear authors,

Thank you for resubmitting your manuscript. I apologize for my probably vague instructions regarding the correlation analysis last time but it requires adaptation. I would suggest you to consider seeking support from a biostatistician.

What you need to correlate is the relative expression levels of the genes that indicate a contractile phenotype with the relative gene expression levels of the genes that indicate a more synthetic phenotype. You basically do this to support the conclusion that the expression of contractile genes decreases whilst at the same time the expression of synthetic genes increases and to answer the question if their expression is somewhat connected/correlated with one another.

You may require a specific correlation test to do that, as your expression levels might be inversely correlated.

If you can show that it would be a suppotive and strong statistical evidence for your conclusions.

I hope this was helpful.

Best regards.

The reviewer.

Answer: We thank the Reviewer 3 for these clarifications. We believe we have met his/her expectations by establishing correlations between contractile and synthetic markers. This representation is indeed interesting and very useful, as it clearly shows that the expression of contractile markers decreases concomitantly with the increase in the expression of synthetic markers. We applied a Pearson test, which shows that these correlations are statistically significant (p-values and correlation coefficients (r) are shown in the corresponding figure).

This new figure has been added in supplemental data file and has also been mentioned in the text.

---

## [Editor Report · Decision Letter 3]

2 Dec 2025

Evidence for loss of contractile phenotype of the mouse aortic vascular smooth muscle (MOVAS) cell line with increasing number of passages in vitro

PONE-D-25-33029R3

Dear Dr. JAISSON,

We’re pleased to inform you that your manuscript has been judged scientifically suitable for publication and will be formally accepted for publication once it meets all outstanding technical requirements.

Kind regards,

Gianfranco Pintus, MSc, PhD.

Academic Editor

PLOS ONE
---

## [Editor Report · Acceptance letter]

PONE-D-25-33029R3

PLOS One

Dear Dr. JAISSON,

I'm pleased to inform you that your manuscript has been deemed suitable for publication in PLOS One. Congratulations! Your manuscript is now being handed over to our production team.

Kind regards,

on behalf of

Dr. Gianfranco Pintus

Academic Editor

PLOS One